# Clinical Investigation of Bioelectric Toothbrush for Dentin Hypersensitivity Management: A Randomized Double-Blind Study

**DOI:** 10.3390/bioengineering11090923

**Published:** 2024-09-14

**Authors:** Hyun-Kyung Kang, Yu-Rin Kim, Ji-Young Lee, Da-Jeong Kim, Young-Wook Kim

**Affiliations:** 1Department of Dental Hygiene, Silla University, 140 Baegyang-daero 700beon-gil, Busan 46958, Republic of Korea; dbfls1712@silla.ac.kr (Y.-R.K.); leeji1021@nate.com (J.-Y.L.); ttakjjung@gmail.com (D.-J.K.); 2ProxiHealthcare Advanced Institute for Science and Technology (PAIST), Seoul 04513, Republic of Korea; ywkim@proxihealthcare.com

**Keywords:** bioelectric toothbrush, dentin hypersensitivity, visual analogue scale (VAS), Schiff Cold Air Sensitivity Scale (SCASS), fluoride toothpaste

## Abstract

Background: The objective of this study was to evaluate how effectively the bioelectric toothbrush can alleviate dentin hypersensitivity (DHS) by using electrostatic forces to remove biofilm from the tooth surface. Methods: This study divided inpatients of a preventative dental clinic between March and October 2023 into the following two groups: a bioelectric toothbrush group (BET, n = 25) and a non-bioelectric toothbrush group (NBET, n = 18) as a control group. This was a randomized double-blind, placebo-controlled trial study. A survey, the number of hypersensitive teeth, the O’Leary index, the visual analogue scale (VAS), and the Schiff Cold Air Sensitivity Scale (SCASS) were also investigated. Results: When fluoride toothpaste was applied with a bioelectric toothbrush, the subjects’ VAS and SCASS scores reflecting symptoms of hyperesthesia significantly decreased over time, as did the number of hypersensitive teeth and the O’Leary index. Moreover, the bioelectric toothbrush was confirmed to be effective in removing dental plaque. Conclusions: Dental clinics must actively promote bioelectric toothbrushes and fluoride toothpaste for patients suffering from hyperesthesia and pain. Furthermore, these items can be suggested as preventative oral care products to patients with potential hyperesthesia.

## 1. Introduction

Dentin hypersensitivity (DHS) is a common condition with a prevalence of 25–30% in the adult population [1], characterized by short, sharp pain due to exposed dentin. Despite advances in understanding DHS, differences in patient selection and diagnostic techniques across studies result in a wide variation in reported prevalence [2,3]. The DHS occurs primarily in exposed dentin, and common causes of dentin exposure include improper brushing, gingival recession, tooth removal during restorative treatment, poor dietary habits, anatomical factors, and periodontal therapy [4,5,6]. The DHS is an ongoing clinical problem that presents a significant challenge to oral professionals and affects patients’ quality of life. The impact of DHS on daily life is a very strong motivator for seeking dental care. These patients tend to be very concerned with their inability to enjoy their favorite foods and drinks [7,8]. The most convincing theory of DHS is the hydrodynamic theory, which explains that irritation to the dentin increases the rate of movement of dental fluid in the dentinal tubules, resulting in excitation of nerve endings in the dentinal tubules or in the outer layer of the pulp, causing pain [9]. The theory suggests that dentin at DHS sites is highly permeable due to the exposure of dentinal tubules [10] and that DHS sites contain dentinal tubules with larger diameters and greater permeability [11]. Reducing or completely eliminating the permeability of the dentinal tubules should reduce or eliminate DHS [10]. However, the hydrodynamic theory does not explain all pain associated with hyperesthesia and does not lead to effective treatment. In particular, diagnostic difficulties lead to delays in treatment and unnecessary suffering for patients. Moreover, while DHS is the most common problem faced by dental practices, there is a lack of universally accepted guidelines for differential diagnosis and reliable treatment selection. The DHS treatment aims to reduce or close the diameter of the exposed dentinal tubules to inhibit the movement of dental tubular fluid and reduce its permeability, thereby relieving perceptual hypersensitivity. To alleviate DHS, research has been conducted on non-invasive treatments involving the application of desensitization agents. In particular, researchers have tried destroying the odontoblast fibrils of the exposed dentinal tubules and blocking the entrance to the tubules [12]. There are numerous preparations, materials, and products on the market for the desensitization treatment of DHS. A recent systematic meta-analysis [13,14] showed that most chemical or physical obstructions of the dentinal tubules significantly reduced DHS symptoms. Additionally, fluoride applications, including acidulated sodium fluoride, sodium silicofluoride, and stannous fluoride, have been recommended, with fluoride iontophoresis being used to enhance the effectiveness of these applications [15,16]. Recently, an electric toothbrush that removes biofilm using electromagnetic waves with electrical force was invented. This bioelectric toothbrush separates biofilm from the dental surface in a non-invasive way using a microcurrent that is harmless to the human body. Simultaneously applying direct and alternating current mechanisms, the bioelectric toothbrush [17] shows an excellent sterilizing effect on biofilm with a minimal current. The direct current continuously applies electrostatic force in one direction and causes fine hydrolysis in the medium, separating the microbial film from the surface. On the other hand, the alternating current consistently induces a vibration with a certain frequency, which effectively increases the permeability of the microbial film and removes it. Therefore, it is believed that the bioelectric toothbrush can be used as an effective device for biofilm removal. Furthermore, along with the saliva and fluoride toothpaste reaction, the bioelectric toothbrush can provide a microcurrent to the teeth that efficiently closes the dentinal tubules. This study aims to determine the effectiveness of the bioelectric toothbrush in relieving DHS by detaching biofilm on the tooth surface based on electrostatic force and promoting the remineralization of fluoride toothpaste. Current non-invasive treatments for DHS have certain limitations. This study aims to examine whether the bioelectric toothbrush can overcome these limitations. In this study, we focused on investigating the extent to which a toothbrush equipped with microcurrent technology can help alleviate symptoms of dentin hypersensitivity.

## 2. Materials and Methods

### 2.1. Ethical Approval and Consent to Participate

The study was approved by the Institutional Review Board of Silla University (1041449-202212-HR-004, Busan, Republic of Korea) and registered as a clinical trial on the WHO International Clinical Trial Registry Platform (ICTRP) (registration date: 29 December 2023, registration number: KCT0009089; https://cris.nih.go.kr/cris/search/detailSearch.do/26158). Additionally, this study was conducted in accordance with the International Council for Harmonization of Technical Requirements for Pharmaceuticals for Human Use (ICH) guidelines. All pertinent information (purpose, procedures, and risks) of this study were explained to all participants. Participants were free to withdraw from the study at any time. Informed consent was provided to all participants prior to enrollment in the clinical trial.

### 2.2. Study Participants

#### 2.2.1. Determination of Number of Subjects Required

The number of subjects required was determined with the G*Power 3.1 program [18], and was confirmed with the repeated measures F test conducted three times. Cohen [19] classified effect sizes as small (0.2), medium (0.5), and large (0.8). For this study, the effect size was set referencing the medium effect size commonly used in G*Power (approximately 0.25 to 0.3). This estimate is considered conservative, something that is often used in exploratory research. Determine was set to Partial 0.06 in Direct, Effect size f was set to 0.25, and α err prob was set to 0.05. As a result of setting Power (1-β err prob) to 0.95 and non-sphericity correction (ε) to 1, the number of participants required was shown to be 36. According to the G*Power [18] analysis, 36 participants are sufficient to achieve 95% power at the given effect size and alpha level. Additionally, since the study employed repeated measures ANOVA, the risk of underestimating the sample size in a structure where the same participants are measured multiple times is reduced. Considering possible eliminations, this study recruited 46 subjects with 10 or more eligible teeth. Participants who understood the overall content of the study and voluntarily agreed to participate were recruited. Subjects were informed that unnecessary personal identifiers, names, and social registration numbers would not be published and would be stored for privacy and confidentiality reasons.

#### 2.2.2. Subject Inclusion and Exclusion Criteria

The inclusion criteria for this study were as follows: patients who were maintaining their periodontal health or seeking preventative treatment (attending the clinic for primary or secondary periodontitis, gingivitis, or early periodontitis diagnosis); without cavities or fractured teeth; patients with at least 10 eligible teeth; who met the following inclusion criteria were recruited: at least two sensitive teeth with a response ≥ 2 on the 10 cm length visual analogue scale (VAS) (0-no pain and 10-unbearable pain) after tactile and evaporative stimulation and/or the presence of a non-carious lesion up to 2 mm deep [20], and/or class I gingival recession [21]; who could be recalled more than four times; who were over 20 years old (adults); who were physically fit; and who fully understood the study and provided written consent. The exclusion criteria were as follows: patients with fewer than 10 suitable teeth; acute toothache; other acute dental infections or pulpitis; severe caries requiring treatment; current antibiotic treatment, long-term medication, or analgesia; allergies or any other undesirable reaction history; a disease or medication that affects inflammation; a compromised immune system (including anti-inflammatory medications); alcohol abuse; pregnancy or breast-feeding; fractured or cracked teeth; pain from whitening or surgery; a defective prosthesis; and enamel hypoplasia.

#### 2.2.3. Study Design and Protocol

In this randomized double-blind placebo-controlled trial study, inpatients of a preventative dental clinic between March and October 2023 who agreed to participate after a dental hygienist with more than 10 years of experience directly explained the study purpose and content were selected as subjects. The experimental and placebo toothbrushes were manufactured to be identical in appearance, ensuring that neither the researchers nor the participants could discern which type of toothbrush was being used. This procedure was implemented to preserve the study’s objectivity. As a result, a total of 46 subjects were recruited. The subjects were then randomly divided into a bioelectric toothbrush group (BET, n = 26) and a non-bioelectric toothbrush group (NBET, n = 20) in a double-blind manner. The way the groups were divided was through a lottery, and 30 chits from Group A and 30 chits from Group B were kept in a box. When participants were asked to choose one of the chits, they were given a toothbrush and toothpaste corresponding to group A or group B. Co-researcher 2 blindfolded the participant by removing all labels from the toothbrush and replacing BET with A and NBET with B. Co-researcher 2 provided the participant with a toothbrush and toothpaste after the participant chose a chit. Co-researcher 1 maintained randomization records. The principal investigator who recorded clinical data also remained blinded to the assignments. It was only after statistical analysis that it was revealed that group A was BET and group B was NBET.

As for the bioelectric toothbrush [17], a toothbrush producing a 100 mA microcurrent with the simultaneous application of direct and alternating current electrical mechanisms was used (Figure 1). For the control group, a toothbrush that did not produce a microcurrent, but emitted light was used, as it appeared superficially identical to the bioelectric toothbrush. During a 7-week period, 3 participants (overseas business trip, moving) were eliminated, resulting in 25 in the BET and 18 in the NBET (Figure 2).

#### 2.2.4. Clinical Examination

To ensure consistency of the oral environment, the participants received an oral examination by a dentist and light scaling by a trained dental hygienist at a preventative dental clinic prior to the study. After a 1-week recovery period, the study commenced. Participants were provided with F 1450 toothpaste and requested to maintain their usual dental care habits, including toothbrushing method and oral health behavior, during the study period. During the study period, intake of carbonated drinks, alcohol, sweet foods, acidic foods (fruits with high acid content such as lemon), and cold water (ice) that could affect the symptoms of cold was limited. On the first day of the study, a survey and an oral examination were conducted (i.e., baseline data were collected), and data were collected a total of 3 times during the 6 weeks of home care (Figure 3).

### 2.3. Variables

#### 2.3.1. Survey

As for the subjects’ sociodemographic characteristics, gender and age were inspected. Furthermore, the presence of hypertension, diabetes, and osteoporosis was confirmed to check for systemic disease. Moreover, education level, marital status, and occupation were investigated as well as smoking and alcohol consumption habits.

#### 2.3.2. Observation Test

As for the oral examination, the O’Leary index [22] was used to confirm the degree of dental plaque. All teeth in the oral cavity were discolored with a tooth surface discolorant, and four tooth surfaces (mesial, distal, buccal, and lingual) were checked. The degree of adhesion (%) was calculated using the O’Leary index, where 1 point is given if the plaque is attached to the tooth surface and 0 points if it is not attached. Furthermore, the Visual Analogue Scale (VAS) [23] was employed to investigate the pain from sensitivity. The VAS is a 10-point scale in which patients indicate their subjective level of pain, with a higher score indicating more severe symptoms. According to the symptoms of sensitivity, the following seven components were measured: when brushing teeth (VAS_brush); when consuming cold water (VAS_water); when drinking carbonated drinks (VAS_carb); when consuming sour food (VAS_sour); when the tooth surface is touched (VAS_ct); when the tooth is exposed to air (VAS_air); and when eating ice (VAS_ice). The Schiff Cold Air Sensitivity Scale (SCASS) [24] refers to the dentist’s report on patients’ symptoms evaluated by other dentists. The documentation is recorded as follows: 0 points for “no response”; 1 point for “there is a response but no request for discontinuation”; 2 points for “discontinuation request and flinching when stimulated”; and 3 points for “requests to discontinue and avoids the intervention due to pain”. Three components of the SCASS were measured. The SCASS_ct was evaluated by a 1–2 s stimulus on the tooth surface at an angle of 45 degrees to the long axis of the tooth using a pressure-sensitive probe (axe pressure-sensitive probe, Bluedent India, Tamil Nadu, India). As for SCASS_air, 3 s of cold air exposure at a distance of 2–3 mm from the relevant tooth surface using an air syringe from the dental unit chair was used. Additionally, SCASS_ice was measured with 3 s of ice stick contact on the relevant tooth surface. Lastly, the number of teeth with sensitive symptoms confirmed through VAS and SCASS was recorded. To increase the internal and external consistency of the evaluator, three simulation experiments were conducted prior to ensure the same measurement value in a subject. Evaluation for the various stimuli was performed at five-minute intervals between each test. Subjects who gave a minimum score of 3 in the numeric rating scale (VAS) and a minimum of 2 in SCASS during the baseline examination were recruited for the study.

### 2.4. Statistical Analysis

A significance level of 5% was set for all of the results derived using SPSS 27.0 for Windows (IBM Corp., Armonk, NY, USA). A frequency analysis of the participants’ demographic characteristics was conducted, and each group’s sensitivity symptoms over time were compared by repeated measures analysis. The correspondence rates between the evaluators for each session were significantly consistent (i.e., 0.871, 0.856, and 0.919 for the VAS, SCASS, and number of teeth with sensitivity, respectively).

## 3. Results

### 3.1. Sociodemographic Characteristics of Two Groups

Table 1 shows the comparison results of the subjects’ sociodemographic characteristics. The proportion of females was higher in both groups, but the average age was higher in BET compared to NBET. The proportions of married individuals in terms of marital status and employed individuals in terms of employment status were the highest in both groups. As for education level, those with undergraduate degrees made up the highest proportion of NBET, while BET had the same proportion of high school graduates and individuals with undergraduate degrees. In both groups, there were higher proportions of people without any of the three systemic diseases and non-smokers. Additionally, frequent alcohol consumption was prevalent in NBET, while occasional alcohol consumption was most common in BET. There was no significant difference in any of the variables, and thus, the homogeneity of the two groups was ensured (*p* > 0.05).

### 3.2. Differences in VAS over Time between the Two Groups

The results of the analysis of the differences in the subjective pain scale (VAS) over time between the two groups are shown in Table 2. All seven sub-components of the VAS showed decreases in pain compared to baseline over time measured at the 1st, 2nd, and 3rd data collection points. Differences between the two groups after the passage of a certain amount of time were observed in VAS_water, VAS_carb, and VAS_ct (*p* < 0.05). No significant difference was noted in any of the components regarding the passage of time or the interaction between the two groups. When the differences between baseline and the 3rd data collection point were investigated, VAS_brush was 2.45 in NBET and 2.84 in BET, showing a larger decrease in BET. Similarly, VAS_water was 2.78 in NBET and 2.92 in BET, and VAS_carb was 1.34 and 1.36 in NBET and BET, respectively. The VAS_sour was 1.22 in NBET and 1.98 in BET, VAS_ct was 0.83 in NBET and 1.96 in BET, VAS_air was 3.11 in NBET and 3.28 in BET, and VAS_ice was 2.11 and 3.64 in NBET and BET, respectively. A significant decrease was observed in VAS_tot, which combines the pain scales of all subjects (18.76 in BET and 14.11 in NBET).

### 3.3. Differences in SCASS over Time between the Two Groups

The results of the analysis of the differences in SCASS in the two groups over time are shown in Table 3. All three sub-components showed decreases in pain compared to baseline over time measured at the 1st, 2nd, and 3rd data collection points. A difference between the two groups after the passage of a certain amount of time was observed in SCASS_ct. Furthermore, significant differences in the passage of time and the interaction between the two groups were noted in SCASS_ice and SCASS_tot (*p* < 0.05). The differences observed compared to baseline at the 3rd data collection point were as follows. The SCASS_ct was 0.83 in NBET and 0.56 in BET, while SCASS_air was 1.32 in BET and 1.12 in NBET. Similarly, SCASS_ice was 1.06 in NBET and 1.72 in BET. A larger decrease in SCASS_tot, which combines the pain scales of all subjects, was observed in BET (3.63) when compared to NBET (3.00).

### 3.4. Differences in the Number of Hypersensitive Teeth and O’Leary Index over Time between the Two Groups

The differences between the two groups in terms of the number of hypersensitive teeth and the O’Leary index over time were observed, and the results are shown in Table 4. Both the number of hypersensitive teeth and the O’Leary index decreased over time from the baseline when measured at the 1st, 2nd, and 3rd data collection points. A difference between the two groups was observed in the O’Leary index after the passage of a certain amount of time, whereas a significant difference between the passage of time and an interaction between the two groups were observed in the number of hypersensitive teeth (*p* < 0.05). As a result of observing the differences compared to baseline at the 3rd data collection point, a larger decrease in the number of hypersensitive teeth was observed in BET (12.52) when compared to NBET (5.05). Similarly, the O’Leary index decreased more in BET (61.82) compared to NBET (8.35).

## 4. Discussion

The DHS is a condition caused by the exposure of dentinal tubules, and it is commonly seen in patients of all ages, except infants. There have been several hypotheses to explain this observation, but clear reasons behind this phenomenon remain unknown [25]. To relieve hyperesthesia symptoms, the movement of liquid through the dentinal tubules must be reduced or blocked, but there has been limited research on using microcurrents to do so. The purpose of this study is to investigate the difference in the effectiveness of relieving DHS between bioelectric toothbrushes and non-bioelectric toothbrushes by applying fluoride detergent. In the VAS, patients self-assess and document their subjective levels of pain or relief. When the differences between the two groups over time were investigated, the pain level of all seven sub-components decreased compared to baseline over time measured at the 1st, 2nd, and 3rd data collection points. Among them, differences between the two groups after the passage of a certain amount of time were shown in VAS_water, VAS_carb, and VAS_ct (*p* < 0.05). This reduction showed a greater effect in BET than in NBET, which appears to be the result of the bioelectric toothbrush improving the sealing ability of dentinal tubules in response to fluoride detergents. Fluoride is known to be effective in alleviating hyperesthesia and remineralizing the tooth surface [26]. Furthermore, the bioelectric toothbrush separates biofilm from the tooth surface with a special 100 mA microcurrent, which is harmless to the human body [17,27]. In other words, based on this evidence, it can be said that the combined effect of the fluoride-containing toothpaste and the bioelectric toothbrush increased the effect of the microcurrent, which relieved subjects’ subjective pain.

The SCASS refers to a document evaluated and recorded by a dentist based on a patient’s symptoms. As a result of examining the differences in the two groups over time, decreased pain was observed in all sub-components compared to baseline at the 1st, 2nd, and 3rd data collection points. Among them, SCASS_ct showed a difference between the two groups after the passage of a certain amount of time. Furthermore, a significant difference in the passage of time and an interaction between the two groups were observed in SCASS_ice and SCASS_tot (*p* < 0.05). As in the VAS, it can be said that the hyperesthesia pain was alleviated over time in the SCASS. When using a hypersensitivity-relieving toothpaste, a smear layer is formed, which allows the active ingredient in the toothpaste to penetrate and close the dentinal tubules. The smear layer formed then increased the penetration of the active ingredient [28]. With the use of a bioelectric toothbrush, which induces a bio-friendly microcurrent to remove biofilm, a bio-friendly electromagnetic wave effect is seen due to electrochemical environmental changes. As a result, the fluoride penetrates deeper into the dentinal tubules of hypersensitive teeth, which results in improved hyperesthesia by blocking or reducing fluid movement within dentinal tubules with increased secondary dentin formation.

When the differences over time between the two groups in the number of hypersensitive teeth and the O’Leary index were investigated, both showed decreases compared to baseline at the 1st, 2nd, and 3rd data collection points. Among them, the O’Leary index showed differences between the two groups after the passage of a certain amount of time. In comparison, the number of hypersensitive teeth showed a significant difference in the passage of time and an interaction between the two groups (*p* < 0.05). This result is similar to that of a study on dental plaque removal using a bioelectric toothbrush [29,30]. A bioelectric toothbrush applies both direct and alternating current mechanisms simultaneously and has optimized electromagnetic wave characteristics, which enables microbial film separation even with a microscopic level of current. It is likely that this characteristic allows plaque removal from the areas where a regular toothbrush cannot reach. It is believed that effective biofilm or plaque removal can assist in the prevention of dental caries, gingivitis, and periodontitis by limiting the bacterial growth in the oral cavity through biofilm separation. A study conducted by Kim and Jang [31] also verified the influence of a bioelectric toothbrush on the fluoride concentration on the tooth enamel surface. According to the study, electron microscopy revealed slightly irregular spherical crystals (fluorapatite) appearing on the enamel surface after participants used a bioelectric toothbrush. This finding shows that bioelectric toothbrushes effectively prevent cavities by increasing fluorapatite formation on the enamel. Therefore, it is believed that the combined use of a bioelectric toothbrush and fluoride-containing toothpaste has several positive effects, including prevention of dental caries, remineralization promotion in early dental caries, and tooth hypersensitivity alleviation. During the study period, no discomfort or side effects were observed among the subjects when using the bioelectric toothbrush and fluoride toothpaste, which suggests the application to be useful not only in clinical settings but also at home.

## 5. Limitations and Suggestions

Despite the excellent effect in alleviating hypersensitivity in this study, it is difficult to generalize its results due to its cross-sectional nature. This study was designed to focus on confirming the possibility as the first exploratory clinical trial to determine the relationship between bioelectric current and DHS. Therefore, in the future, we plan to conduct clinical verification and mechanism research on the synergy effect of fluoride and bioelectric current to improve DHS, and conduct research to confirm DHS improvement according to changes in toothpaste ingredients. Additionally, because our study is limited by the relatively short period of 7 weeks and the small number of subjects, we need to be careful about generalizing and interpreting our results. Accordingly, future studies are planned to include long-term follow-up investigations with a larger sample size. Despite these limitations, we believe that our study is meaningful enough to provide basic data as the first clinical study applying a bioelectric toothbrush to DHS.

## 6. Conclusions

This study was an exploratory clinical study to confirm the possibility of future research and confirmed the positive effect of bioelectric toothbrushes on DHS. Therefore, it will be possible to suggest bioelectric toothbrushes as a home care oral care product to patients and potential patients visiting with DHS.

## Figures and Tables

**Figure 1 bioengineering-11-00923-f001:**
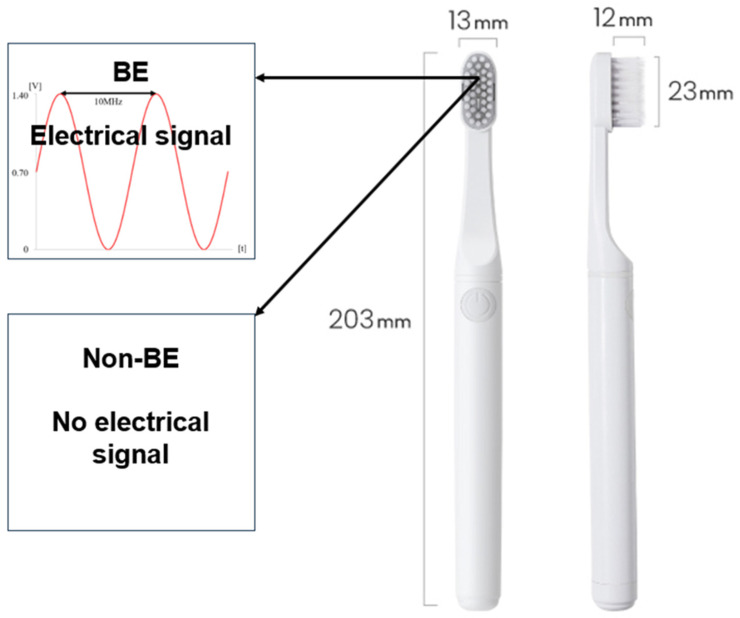
Schematics of the BE and non-BE toothbrush (ProxiHealthcare, Seoul, Republic of Korea).

**Figure 2 bioengineering-11-00923-f002:**
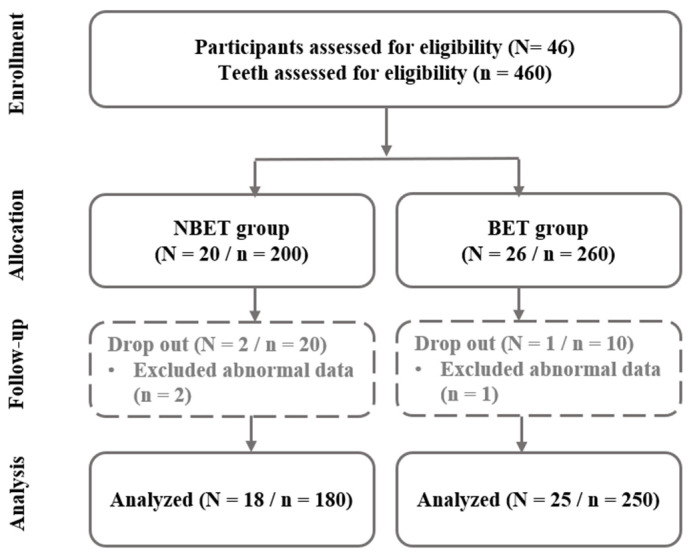
Flowchart of participants.

**Figure 3 bioengineering-11-00923-f003:**
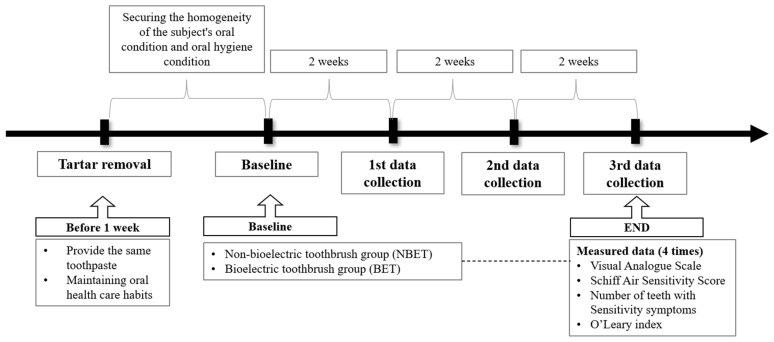
Flowchart of study method.

**Table 1 bioengineering-11-00923-t001:** Sociodemographic characteristics of two groups.

Characteristics		NBET (N = 18/n= 180)	BET (N = 25/n = 250)	* *p*-Value
Gender	Male	5 (27.8)	7 (28.0)	1.000
	Female	13 (72.2)	18 (72.0)	
^￥^ Age (mean ± SD)		42.50 ± 15.10	51.00 ± 13.81	0.062
Marital status	Single	6 (33.3)	5 (20.0)	0.220
	Married (spouse)	11 (61.1)	20 (80.0)	
	Married (non-spouse)	1 (5.6)	0 (00.0)	
Employment status	Unemployed	4 (22.2)	11 (44.0)	0.199
	Employed	14 (77.8)	14 (56.0)	
Level of education	≤Middle school	1 (5.6)	3 (12.0)	0.226
	High school	4 (22.2)	11 (44.0)	
	≥Undergraduate	13 (72.2)	11 (44.0)	
High blood pressure	No	15 (83.3)	21 (84.0)	1.000
	Yes	3 (16.7)	4 (16.0)	
Diabetes	No	16 (88.9)	24 (96.0)	0.562
	Yes	2 (11.1)	1 (4.0)	
Osteoporosis	No	17 (94.4)	21 (84.0)	0.380
	Yes	1 (5.6)	4 (16.0)	
Smoking status	Non-smoker	11 (61.1)	20 (80.0)	0.308
	Former smoker	4 (22.2)	2 (8.0)	
	Smoker	3 (16.7)	3 (12.0)	
Alcohol consumption	Non-drinker	6 (33.3)	5 (20.0)	0.284
	Occasional drinker	5 (27.8)	13 (52.0)	
	Frequent drinker	7 (38.9)	7 (28.0)	

^￥^ *p*-values are determined by independent *t*-test analysis, * *p*-values are determined by Fisher’s exact test (*p* < 0.05), Values are means ± standard deviations, Non-bioelectric toothbrush group; NBET, bioelectric toothbrush group; BET.

**Table 2 bioengineering-11-00923-t002:** Differences in VAS over time between the two groups.

Variable	Group	Baseline	1st	2nd	3rd	Source	F	* *p*-Value
VAS_brush	NBET	4.39 ± 2.25	3.28 ± 1.84	2.56 ± 1.58	1.94 ± 1.69	Group	1.726	0.196
	BET	4.04 ± 2.99	2.68 ± 2.45	1.56 ± 2.26	1.20 ± 1.58	Time	21.276	**<0.001**
						Group * Time	0.288	0.777
VAS_water	NBET	6.00 ± 1.57	4.89 ± 1.99	3.56 ± 1.54	3.22 ± 1.73	Group	7.028	**0.011**
	BET	4.40 ± 2.93	3.92 ± 2.31	2.24 ± 2.40	1.48 ± 1.78	Time	33.514	**<0.001**
						Group * Time	0.563	0.604
VAS_carb	NBET	3.67 ± 2.30	3.39 ± 2.15	2.78 ± 2.13	2.33 ± 2.14	Group	13.306	**<0.001**
	BET	2.32 ± 2.67	1.96 ± 2.30	1.12 ± 1.62	0.96 ± 1.57	Time	7.291	**0.010**
						Group * Time	0.104	0.912
VAS_sour	NBET	4.00 ± 2.52	3.22 ± 2.56	3.00 ± 2.38	2.78 ± 2.26	Group	3.556	0.066
	BET	3.32 ± 3.24	2.68 ± 2.68	1.40 ± 1.68	1.24 ± 2.03	Time	6.993	**0.001**
						Group * Time	0.948	0.399
VAS_ct	NBET	3.83 ± 2.79	3.33 ± 2.66	3.50 ± 2.89	3.00 ± 2.79	Group	10.972	**0.002**
	BET	2.68 ± 3.07	1.16 ± 2.29	1.64 ± 2.36	0.72 ± 1.21	Time	3.593	**0.027**
						Group * Time	0.662	0.533
VAS_air	NBET	6.83 ± 1.51	5.17 ± 2.36	4.33 ± 2.54	3.72 ± 2.63	Group	1.562	0.218
	BET	6.12 ± 2.19	4.52 ± 2.97	3.76 ± 2.28	2.84 ± 2.43	Time	23.545	**<0.001**
						Group * Time	0.055	0.983
VAS_ice	NBET	6.50 ± 2.26	5.17 ± 2.77	4.39 ± 2.36	3.89 ± 2.74	Group	0.010	0.921
	BET	7.44 ± 2.36	5.36 ± 2.50	3.80 ± 2.81	3.08 ± 2.84	Time	34.739	**<0.001**
						Group * Time	2.343	0.076
VAS_tot	NBET	35.00 ± 8.68	28.44 ± 10.80	24.11 ± 11.52	20.89 ± 11.46	Group	6.069	**0.018**
	BET	30.28 ± 15.42	22.28 ± 12.85	15.56 ± 11.45	11.52 ± 10.06	Time	31.667	**<0.001**
						Group * Time	0.709	0.497

* *p*-values are determined by repeated measures ANOVA test (*p* < 0.05).; Values are means ± standard deviations; significant (bold). Mauchly’s sphericity test; *p* < 0.05 (*p*-value = Greenhouse–Geisser), Non-bioelectric toothbrush group; NBET, bioelectric toothbrush group; BET.

**Table 3 bioengineering-11-00923-t003:** Differences in SCASS over time between the two groups.

Variable	Group	Baseline	1st	2nd	3rd	Source	F	* *p*-Value
SCASS_ct	NBET	1.33 ± 0.97	1.11 ± 0.96	0.94 ± 0.80	0.50 ± 0.62	Group	9.762	**0.003**
	BET	0.76 ± 0.88	0.40 ± 0.76	0.44 ± 0.71	0.20 ± 0.41	Time	8.421	**<0.001**
						Group * Time	0.755	0.499
SCASS_air	NBET	2.06 ± 0.80	1.67 ± 0.69	1.22 ± 0.55	0.94 ± 0.73	Group	1.156	0.289
	BET	2.08 ± 0.70	1.36 ± 0.76	0.96 ± 0.94	0.76 ± 0.66	Time	35.843	**<0.001**
						Group * Time	0.679	0.567
SCASS_ice	NBET	2.17 ± 0.79	1.83 ± 1.04	1.56 ± 0.86	1.11 ± 0.76	Group	1.205	0.279
	BET	2.48 ± 0.65	1.54 ± 0.71	1.04 ± 0.79	0.76 ± 0.78	Time	43.006	**<0.001**
						Group * Time	3.967	**0.010**
SCASS_tot	NBET	5.56 ± 1.86	4.61 ± 2.03	3.72 ± 1.67	2.56 ± 1.42	Group	5.041	**0.030**
	BET	5.35 ± 1.77	3.32 ± 1.82	2.44 ± 2.08	1.72 ± 1.46	Time	42.546	**<0.001**
						Group * Time	1.340	0.264

* *p*-values are determined by repeated measures ANOVA test (*p* < 0.05).; Values are means ± standard deviations; significant (bold). Mauchly’s sphericity test; *p* < 0.05 (*p*-value = Greenhouse–Geisser), Non-bioelectric toothbrush group; NBET, bioelectric toothbrush group; BET.

**Table 4 bioengineering-11-00923-t004:** Differences in the number of hypersensitive teeth and O’Leary index over time between the two groups.

Variable	Group	Baseline	1st	2nd	3rd	Source	F	* *p*-Value
Hypersensitive teeth	NBET	9.22 ± 7.34	6.89 ± 6.44	5.44 ± 4.11	4.17 ± 2.92	Group	2.622	0.113
	BET	17.68 ± 11.99	9.36 ± 8.51	5.96 ± 6.42	5.16 ± 6.90	Time	26.049	**<0.001**
						Group * Time	5.676	**0.009**
O’Leary index	NBET	28.16 ± 19.23	23.59 ± 14.17	21.41 ± 14.49	19.81 ± 14.66	Group	6.415	**<0.001**
	BET	45.78 ± 24.17	36.59 ± 20.16	34.04 ± 20.88	28.96 ± 18.79	Time	10.662	**<0.001**
						Group * Time	1.135	0.319

* *p*-values are determined by repeated measures ANOVA test, (*p* < 0.05).; Values are means ± standard deviations.; significant (bold). Mauchly’s sphericity test; *p* < 0.05 (*p*-value = Greenhouse–Geisser), Non-bioelectric toothbrush group; NBET, bioelectric toothbrush group; BET.

## Data Availability

Data Availability Statements are available in section The datasets used and/or analyzed during the current study available from the corresponding author on reasonable request.

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
