# Peer review of "Clinical Investigation of Bioelectric Toothbrush for Dentin Hypersensitivity Management: A Randomized Double-Blind Study"

_bioengineering, 2024, doi:10.3390/bioengineering11090923_

Round 1

Reviewer 1 Report

Comments and Suggestions for Authors

The study is well written and structured, but there are still issues that need to be addressed before the study is ready for publication.

1. Please follow CONSORT’s guidelines for randomized clinical trials. It should be stated in the title that the study is a randomized clinical trial. Also for the abstract, also follow CONSORT’s guidelines for abstracts of RCTs: https://www.equator-network.org/wp-content/uploads/2011/10/CONSORT-for-Abstracts.pdf

2. Structure the introduction better with clear and distinct paragraphs which explain a particular aspect of the introduction.

3. This sentence “A better understanding of the DHS burden and associated factors can help to plan resources to reduce or prevent the discomfort caused by this condition and aid in the decision-making process” seems excessive for your study subject since after that you explain some of the possible factors associated with it.
4. In methods and materials, under ethical…. “All methods were carried out in accordance with the relevant guidelines and regulations.”, mention what guidelines and regulations with citations to them.

5. In “2.2.1. Determination of number of subjects required” based on what previous or pilot study the determine was set to Partial 0.06 in Direct, Effect size f was set to 0.2526456? Please add that citation or explain how did you reached this number.

6. In “2.2.3. Study design and protocol” how the randomization was done (randomize table; randomizer.org; excel; etc.) and what was the allocation ratio since one group has a noticeably larger sample size than the other? Please explain.

7. Based on CONSORT guidelines, there is no need to use statistical analyses to compare the distribution of baseline characteristics between subjects in different groups. The table is sufficient.

8. For results tables, there is no such thing as a zero P value, change the 0.000 P values to <0.001.

9. Again, for the discussion, please restructure it into a less wordy and distinct paragraphs.

10. Add a “limitations and suggestions” subsection at the end of the discussion.

Author Response

  1. Please follow CONSORT’s guidelines for randomized clinical trials. It should be stated in the title that the study is a randomized clinical trial. Also for the abstract, also follow CONSORT’s guidelines for abstracts of RCTs: https://www.equator-network.org/wp-content/uploads/2011/10/CONSORT-for-Abstracts.pdf

       Answer: We have revised the title and abstract based on your comments.

  1. Structure the introduction better with clear and distinct paragraphs which explain a particular aspect of the introduction.

     Answer: We have condensed the overly lengthy sections on the general overview of dentin hypersensitivity and focused this study on examining the extent to which a toothbrush equipped with microcurrent technology can help alleviate symptoms of dentin hypersensitivity.

  1. This sentence “A better understanding of the DHS burden and associated factors can help to plan resources to reduce or prevent the discomfort caused by this condition and aid in the decision-making process” seems excessive for your study subject since after that you explain some of the possible factors associated with it.

        Answer: This sentence removed “A better understanding of the DHS burden and associated factors can help to plan resources to reduce or prevent the discomfort caused by this condition and aid in the decision-making process”

  1. In methods and materials, under ethical…. “All methods were carried out in accordance with the relevant guidelines and regulations.”, mention what guidelines and regulations with citations to them.

        Answer: Based on your views, I have added more detail to the section. thank you.

  1. In “2.2.1. Determination of number of subjects required” based on what previous or pilot study the determine was set to Partial 0.06 in Direct, Effect size f was set to 0.2526456? Please add that citation or explain how did you reached this number.

         Answer: Effect size in the given data: An effect size of f=0.25 is a standardized indicator of how much an independent variable (such as the difference between groups) influences a dependent variable (such as the measured outcome) in a study. In statistical tests like ANOVA (Analysis of Variance), the effect size represents the magnitude of the “true difference” that the researcher aims to detect, reflecting the expected difference under the study's conditions.

Significance of an effect size of f=0.25: Medium-sized effect: According to Cohen's criteria, f=0.25 corresponds to a medium effect size. This indicates that there is a noticeable difference between the groups, though not a very large one. It can be interpreted as a difference that may have clinical or practical significance.

Relationship to sample size: To detect a medium effect size, a sample size of 36 participants was calculated as necessary. This calculation is based on an α error probability of 0.05 and a power of 0.95, ensuring that the study is sufficiently powered to detect a meaningful difference.

Role in study design: Establishing an effect size during the study design phase is crucial. If the effect size is set too small, it may require an unnecessarily large sample size; if set too large, it may fail to detect a meaningful difference. Therefore, setting f=0.25 represents a reasonable compromise, ensuring that the study results are likely to have clinical or practical relevance.

In conclusion, an effect size of f=0.25 was set in this study to detect a medium-sized difference between groups. This plays a critical role in ensuring the reliability and power of the study, and it is a key factor in determining the necessary sample size to detect statistically significant differences.

  1. In “2.2.3. Study design and protocol” how the randomization was done (randomize table; randomizer.org; excel; etc.) and what was the allocation ratio since one group has a noticeably larger sample size than the other? Please explain.

       Answer: We have added a detailed description of the randomization method via lottery.

  1. Based on CONSORT guidelines, there is no need to use statistical analyses to compare the distribution of baseline characteristics between subjects in different groups. The table is sufficient.

       Answer: e present Table 1 to illustrate the homogeneity of the two groups. thank you.w

  1. For results tables, there is no such thing as a zero P value, change the 0.000 P values to <0.001.

      Answer: We've made the changes you mentioned.

  1. Again, for the discussion, please restructure it into less wordy and distinct paragraphs.

      Answer: We have modified the content according to your views.

  1. Add a “limitatiower:ns and suggestions” subsection at the end of the discussion.

      Answer: We have added a [Limitations and Suggestions] section under Discussion based on your views.

Reviewer 2 Report

Comments and Suggestions for Authors

With only 25 participants in the bioelectric toothbrush group and 18 in the control group, the sample size is quite limited and the 7 day study period is too short. This reduces the statistical power of the study and limits the generalizability of the findings. The effect size (f=0.2526456f = 0.2526456) was determined, but the rationale behind this specific value isn't provided. Effect size is crucial for powering a study, and its selection should be based on prior studies or pilot data. Three participants were eliminated during the 7-day period, reducing the final number to 25 in the BET group and 18 in the NBET group. The reasons for dropout aren’t specified, which could be important for understanding potential biases. The findings are difficult to interpret and apply in broader clinical contexts.

Author Response

Answer:

In the sentence, “Three participants were eliminated during the 7-day period,” "7-day" was a typo. It has been corrected and reinserted.

Effect size in the given data: An effect size of f=0.25 is a standardized indicator of how much an independent variable (such as the difference between groups) influences a dependent variable (such as the measured outcome) in a study. In statistical tests like ANOVA (Analysis of Variance), the effect size represents the magnitude of the “true difference” that the researcher aims to detect, reflecting the expected difference under the study's conditions.

Significance of an effect size of f=0.25: Medium-sized effect: According to Cohen's criteria, f=0.25 corresponds to a medium effect size. This indicates that there is a noticeable difference between the groups, though not a very large one. It can be interpreted as a difference that may have clinical or practical significance.

Relationship to sample size: To detect a medium effect size, a sample size of 36 participants was calculated as necessary. This calculation is based on an α error probability of 0.05 and a power of 0.95, ensuring that the study is sufficiently powered to detect a meaningful difference.

Role in study design: Establishing an effect size during the study design phase is crucial. If the effect size is set too small, it may require an unnecessarily large sample size; if set too large, it may fail to detect a meaningful difference. Therefore, setting f=0.25 represents a reasonable compromise, ensuring that the study results are likely to have clinical or practical relevance.

In conclusion, an effect size of f=0.25 was set in this study to detect a medium-sized difference between groups. This plays a critical role in ensuring the reliability and power of the study, and it is a key factor in determining the necessary sample size to detect statistically significant differences.

This study was designed and included in the review as the first exploratory clinical trial to investigate the relationship between microcurrents and dental hypersensitivity symptoms.

Our research roadmap begins with confirming the feasibility of improving tooth sensitivity symptoms in this study, which serves as the first step. In the future, we plan to expand into various studies. Potential future research topics include the clinical validation of the synergistic effect of microcurrents with fluoride in improving dental hypersensitivity symptoms, investigations into the underlying mechanisms, and the evaluation of symptom improvement by altering toothpaste ingredients.

The study design is informed by existing research and medical statistics, with the purpose of this study being exploratory, aimed at assessing the potential for future research.

Round 2

Reviewer 1 Report

Comments and Suggestions for Authors

The manuscript has been improved by the authors and can be recommended for publication.